# Remediation of Chromium (VI) from Groundwater by Metal-Based Biochar under Anaerobic Conditions

Yating Jiang [1], Min Dai [2], Fei Yang [1], Imran Ali [3,*], Iffat Naz [4,*] and Changsheng Peng [1,2,*]

[1] The Key Lab of Marine Environmental Science and Ecology, Ministry of Education, Department of Environmental Engineering, Ocean University of China, Qingdao 266100, China; jiang1260245363@163.com (Y.J.); yfgarfee@126.com (F.Y.)
[2] Guangdong Provincial Key Laboratory of Environmental Health and Land Resource, Department of Environmental Engineering, Zhaoqing University, Zhaoqing 526061, China; daimin1007@163.com
[3] Department of Environmental Science and Engineering, College of Chemistry and Environmental Engineering, Shenzhen University, Shenzhen 518060, China
[4] Department of Biology, Science Unit, Deanship of Educational Services, Qassim University, Buraidah 51452, Saudi Arabia
* Correspondence: imranali@szu.edu.cn (I.A.); i.majid@qu.edu.sa (I.N.); pcs005@sohu.com (C.P.)

**Abstract:** Iron salt-modified biochar has been widely used to remove Cr(VI) pollution due to the combination of the generated iron oxides and biochar, which can bring positive charge and rich redox activity. However, there are few comprehensive studies on the methods of modifying biochar with different iron salts. In this study, two iron salt ($FeCl_3$ and $Fe(NO_3)_3$) modification methods were used to prepare two Fe-modified biochar materials for removing Cr(VI) in simulated groundwater environment. It was revealed by systematic characterization that $FeCl_3$@BC prepared via the $FeCl_3$ modification method, has larger pore size, higher zeta potential and iron oxide content, and has higher Cr(VI) adsorption-reduction performance efficiency as compared to $Fe(NO_3)_3$@BC prepared via $Fe(NO_3)_3$ modification method. Combined with XRD and XPS analyses, $Fe_3O_4$ is the key active component for the reduction of Cr(VI) to Cr(III). The experimental results have shown that acidic conditions promoted Cr(VI) removal, while competing ions ($SO_4^{2-}$ and $PO_4^{3-}$) inhibited Cr(VI) removal by $FeCl_3$@BC. The Elovich model and intra-particle diffusion model of $FeCl_3$@BC can describe the adsorption behavior of Cr(VI) well, indicating that both the high activation energy adsorption process and intra-particle diffusion control the removal process of Cr(VI). The Freundlich model ($R^2 > 0.999$) indicated that there were unevenly distributed chemisorptions centers on the $FeCl_3$@BC surface. Stability experiments exposed that $FeCl_3$@BC was stable under neutral, acidic, and alkaline conditions. Furthermore, the main mechanisms of $FeCl_3$@BC removal of Cr(VI) include electrostatic adsorption, chemical reduction, ion exchange, and co-precipitation. In conclusion, our findings provide a new insight for the selection of iron salt-modified biochar methods, and will also be beneficial for the preparation of more efficient Fe-modified biochars in the future.

**Keywords:** iron salt; iron oxides; Fe-modified biochar; adsorption and reduction; stability; mechanisms

## 1. Introduction

Chromium (Cr) is one of the heavy metal contaminants, primarily originated from substandard discharges from industries like tanning, electroplating, textile printing, dyeing etc., and causes serious pollution of groundwater [1]. It occurs mainly as trivalent chromium Cr(III) and hexavalent chromium Cr(VI) [2–4], and usually Cr(VI) exists in water as $HCrO_4^-$ and $CrO_4^{2-}$, and has higher solubility, mobility, and toxicity than Cr(III) [5,6]. Moreover, continued exposure to Cr(VI)-contaminated environment can seriously affect human skin, kidney, and respiratory system, etc. Hence, the remediation of Cr(VI) polluted groundwater is a serious problem.

Biochar is a carbon-rich material fabricated via pyrolysis and carbonization of biomass. It is a good environment-friendly material due to its low production cost and no secondary

pollution. Meanwhile, it is a porous medium with huge specific surface area and abundant functional groups, which can be used for the removal of heavy metal pollutants. Cr(VI) is a redox-sensitive heavy metal pollutant, and biochar can remove it through adsorption and reduction [7–11]. It is reported that biochar mainly removes Cr(VI) through electrostatic adsorption, redox, co-precipitation, ion exchange, and other mechanisms [12]. However, surface of original biochar is usually negatively charged, while Cr(VI) usually occurs as an anion, which limits its adsorption ability by biochar [13]. In addition, the limited redox sites on the surface of pristine biochar also limited its ability to reduce Cr(VI). Therefore, metal oxides are usually introduced into the surface of biochar by modification techniques to increase the active site and zero charge point ($pH_{pzc}$), thus enhancing the adsorption and reduction capacity of Cr(VI) [14–16].

Due to the chemical properties of iron itself, the iron oxides generated during oxygen-limited pyrolysis of iron combined with biomass can bring positive charge and rich redox activity, which has been used to modify biochar to promote its application in Cr(VI) remediation [17]. There are many kinds of iron salts, the more common ones are $FeCl_3$ [18,19] and $Fe(NO_3)_3$ [2,20]. Yi et al. [14] modified *Egeria najas* biochar with $FeCl_3$, and found that the modified biochar had lower porosity than the original material carbon, but loaded with $\gamma$-$Fe_2O_3$ and $Fe_3O_4$ crystals, thus improved its reduction ability to Cr(VI). Further, Wang et al. [21] have modified *Enteromorpha prolifera* (EP) biochar with $FeCl_3$ and found that the surface of the modified biochar is loaded with $\gamma$-$Fe_2O_3$ crystals, giving it a higher surface polarity and a larger specific surface, and significantly improving the removal capacity of Cr(VI). Then, Yang et al. [22] prepared active iron-biochar composites (FeBC-1) by co-pyrolytic $FeCl_3$ modification method. The results showed that FeBC-1 has a larger specific surface area and rich iron oxides (FeO, $Fe_2O_3$, and $Fe_3O_4$), and has higher adsorption and reduction activity for Cr(VI). Another research group modified *Melia azedarach* wood biochar with $Fe(NO_3)_3$ and observed that the surface of the modified biochar was loaded with $Fe_3O_4$ crystals, thereby improving its reducing ability to Cr(VI) [20]. In conclusion, the types of iron oxides loaded on biochar will change the physicochemical properties of biochar, thus making the removal effect of Cr(VI) different. Therefore, the selection of iron salts is an important strategy in the preparation of iron oxide-supported biochar materials, which can further expand the application of biochar. Considering the diversity of iron salt modification methods, the concept of iron oxide species has become a very useful principle, although no one has compared the same biomass modified with different iron salts. The present study employed this recent progress to systematically reveal the effects of different iron salts modification methods on the content and types of biochar-supported iron oxides and Cr(VI) removal through comparative experiments and a series of characterizations.

Few studies have compared different methods of modifying wheat straw with iron salts. Therefore, this research was designed to clarify the specific effects of different iron-modified biochars for the Cr(VI) removal. Specifically, this research includes: (1) Fabrication of Fe-modified biochar using two different types of iron salts ($FeCl_3$ and $Fe(NO_3)_3$) to co-pyrolyze; (2) evaluation of the removal performance of Cr(VI) to determine the most suitable iron salts; (3) confirmation of the key role of $FeCl_3$@BC in Cr(VI) elimination; and (4) to clarify the removal mechanism of Cr(VI) removal by $FeCl_3$@BC.

## 2. Materials and Methods

### 2.1. Materials

Various analytically pure reagents such as potassium dichromate ($K_2Cr_2O_7$), ferric chloride ($FeCl_3$), ferric nitrate ($Fe(NO_3)_3$), hydrochloric acid (HCl), sodium hydroxide (NaOH), nitric acid ($HNO_3$), acetone, sulfuric acid ($H_2SO_4$), phosphoric acid ($H_3PO_4$), sodium chloride (NaCl), sodium nitrate ($NaNO_3$), sodium sulfate ($Na_2SO_4$), sodium hydrogen phosphate ($Na_2HPO_4$), 1,5-diphenylcarbazide ($C_{13}H_{14}N_4O$) were used in the current research. The wheat stalks were collected from wheat fields in Linyi City, Shandong, China. After collection, deionized water was used for washing it. Then it was dried at 80 °C in an

oven and, was crushed into a powder. The powder was sieved through a 60-mesh sieve for further application.

## 2.2. Fabrication of Biochar and Metal-Based Biochar

For the fabrication of pristine biochar (BC), the powder of wheat straw was pyrolyzed for 2 h in a muffle furnace, at a heating rate of 10 to 500 °C/ min. For the preparation of metal-based biochars, 28 g of wheat straw powder was added into 100 mL $FeCl_3$ and $Fe(NO_3)_3$ solutions (1 mol/L concentration) and then stirred for 12 h at 60 °C. Thereafter, the obtained product was dried at 80 °C in an oven, and then pyrolyzed for 2 h in a muffle furnace. The $FeCl_3$-modified biochar sample was named $FeCl_3$@BC, and similarly the $Fe(NO_3)_3^-$ modified biochar sample was named $Fe(NO_3)_3$@BC. The as-prepared metal-based biochars were washed by deionized water, then oven dried at 80 °C. Finally, the synthesized material was stored in sealed bags for its future experimentations. The detailed fabrication protocol is illustrated in Figure S1.

## 2.3. Characterization of the Prepared Biochars

To find out the specific surface area of the prepared biochar samples, the Bruanuer–Emmett–Teller adsorption technique (BET-$N_2$, Micromeritics ASAP2020) was employed. Its pore size and volume were determined by the Barrett–Joyner–Halenda (BJH) technique. The scanning electron microscope (SEM-EDS, FEI Quanta 250FEG) was employed to reveal its surface morphology and element composition. Further, Fourier transform infrared spectroscopy (FTIR) provided information about the surface functional groups of the biochars. To explore its crystal structure, X-ray diffraction (XRD, X-ray diffractometer model X pert3, Malvern Panalytical) spectrum was collected in the 2θ range of 10–80°. Further, the surface chemical composition changes of the material were determined by X-ray photoelectron spectroscopy (XPS, ESCALAB Xi+, Thermo Fisher). Finally, to determine $pH_{pzc}$ the zeta potential was provided via a zeta potential analyzer (Zetasizer Nano ZS90, Malvern Panalytical) [14,22].

## 2.4. Batch Experiment

Batch experimentations were performed in 150 mL polyethylene bottles agitated at 150 rpm under 25 °C. To simulate the anaerobic environment of groundwater, the Cr(VI) solution was purged with high-purity $N_2$ to eradicate dissolved oxygen. Then 0.5 g BC, $FeCl_3$@BC, or $Fe(NO_3)_3$@BC was added into 100 mL of 100 mg/LCr(VI) solution respectively to investigate Cr(VI) elimination performance of different materials. Further, different quantities (0.2–0.5 g) of $FeCl_3$@BC were added to 100 mL of 100 mg/L Cr(VI) solution to determine the optimal dosage. For adsorption kinetics and isotherm experiments, 0.2 g $FeCl_3$@BC was used in 100 mL Cr(VI) solutions of different concentrations (60–250 mg/L). In addition, the influence of initial pH such as 3, 5, 7, and 9, and competing ions viz, $NO_3^-$, $SO_4^{2-}$, $HCO_3^-$, and $PO_4^{3-}$ etc., for the Cr(VI) elimination were also measured. The stability of the material was checked by increasing the used samples under neutral (0.1 mol/L NaCl), basic (0.1 mol/L NaOH), and acidic (0.1 mol/L HCl) environments and all experiments were conducted in triplicate. After the reaction, solid–liquid separations were achieved with filter membranes (0.22 μm). Total Cr(VI) and Fe were confirmed by ICP-OES (720ES, Agilent). Finally, the Cr(VI) concentration was estimated by an ultraviolet-visible spectrophotometer at 540 nm by using 1,5-diphenylcarbazide as a reagent.

## 2.5. Kinetics and Isotherm Analyses

The pseudo-first-order (Equation (1)), pseudo-second-order (Equation (2)), Elovich equation (Equation (3)), and intra-particle diffusion model (Equation (4)) were employed to reveal the rate control steps of material transfer and physicochemical reaction in the adsorption process, as follows:

Pseudo-first-order model:

$$\ln(q_e - q_t) = \ln q_e - K_1 t \tag{1}$$

Pseudo-second-order model:

$$q_t = \frac{q_e^2 K_2 t}{1 + q_e K_2 t} \tag{2}$$

Elovich equation:

$$q_e = \frac{1}{\beta} \ln(1 + \alpha\beta t) \tag{3}$$

where, $q_e$(mg/g) and $q_t$ (mg/g) indicate the adsorption capacity of Cr(VI) at equilibrium time and t time respectively. $K_1$(g/mg/h)and $K_2$(g/mg/h) are the corresponding rate constants, $\alpha$ (mg/g/h) and $\beta$ (g/mg) are the initial adsorption rate and desorption ratio constants, respectively.

Intra-particle diffusion model:

$$q_t = K_3 t^{1/2} + C \tag{4}$$

where, $K_3$ (g/mg/h) is constant of internal diffusivity, and $C$ is the intercept representing the boundary layer thickness.

The adsorption mechanism is reflected by adsorption isotherm model and adsorption layer structure through certain constants. Further, the Langmuir model (Equation (5)) accepts uniform adsorbent surface with single layer of adsorption. The Freundlich isotherm (Equation (6)) admits non-uniform models for adsorbent surface, as follows:

Langmuir model:

$$q_e = \frac{K_L q_{max} C_e}{1 + K_L C_e} \tag{5}$$

Freundlich model:

$$q_e = K_F C_e^{1/n} \tag{6}$$

where, $C_e$ is the equilibrium concentration; $q_e$ is the equilibrium adsorption capacity; $q_m$ is the maximum adsorption capacity; $K_L$ is the Langmuir model constant; $K_F$ is Freundlich model constant; and *1/n* is the adsorption strength.

## 3. Results and Discussion

### 3.1. Influence of Fe$^{3+}$ Solution on Cr(VI) Removal by Biochar

It is clearly illustrated in Figure 1, that the Fe$^{3+}$ solution modification method effectively improved the Cr(VI) elimination ability of biochar. It should be noted that the Cr(VI) removal capability of FeCl$_3$@BC is 99.78%, Fe(NO$_3$)$_3$@BC is 21.47%, BC is 15.13%. The results have exposed that both modification methods can improve the removal efficiency of Cr(VI), but different ferric salt-modified biochars have a great influence on the removal efficiency of Cr(VI). It may be due to the different types and contents of iron oxide crystals supported on the biochar surface by the two modification methods [23]. Therefore, it is necessary to conduct asystematic characterization analysis of the three materials (BC, FeCl$_3$@BC, and Fe(NO$_3$)$_3$@BC), so as to clarify the reasons for the difference in Cr(VI) removal of biochar modified with different iron salts.

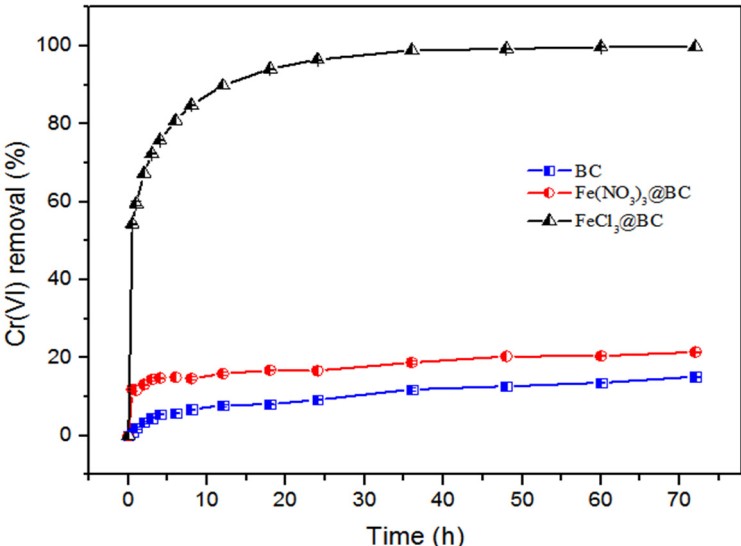

**Figure 1.** The quantity of Cr(VI) eliminated by BC, FeCl$_3$@BC, and Fe(NO$_3$)$_3$@BC.

*3.2. Comparison of the As-Prepared Biochars*

The N$_2$ adsorption–desorption isotherms of BC, FeCl$_3$@BC, and Fe(NO$_3$)$_3$@BC (Figure S2) showed obvious hysteresis loop, which was almost consistent with H4 type, indicating that the three materials all had mesoporous structures [24]. Table 1 illustrated the corresponding surface area and porosity parameters. Compared with BC, the pore size of Fe(NO$_3$)$_3$@BC decreased while the specific surface area increased. This may be because ferric nitrate can promote the release of low-molecular-weight organics in wheat straw during pyrolysis, thus causing changes in the porous structure and specific surface area [25]. In addition, the irregular superposition and dispersion of iron oxide particles formed on the Fe(NO$_3$)$_3$@BC surface during the pyrolysis process resulted in a significant increase in specific surface area, but a large number of iron oxide particles would block the biochar pores and reduce the pore size [22]. Compared with BC, the specific surface area of FeCl$_3$@BC decreased, but had the largest pore size, which is presumably due to the fact that the iron oxide particles loaded on the FeCl$_3$@BC surface block the micropores of the biochar after modification [14]. Studies have shown that large pores were conducive to mass transfer and have a smaller diffusion resistance, which improved the removal ability of Cr(VI) [2,14]. Hence, FeCl$_3$@BC significantly increased the Cr(VI) removal ability.

**Table 1.** The surface area and porosity of non-modified biochar (BC) and metal-based biochar (FeCl$_3$@BC and Fe(NO$_3$)$_3$@BC).

| Sample | BET Surface Area (m$^2$/g) | BJH Desorption of Cumulative Pore Volume (cm$^3$/g) [a] | Average Pore Size (nm) |
|---|---|---|---|
| BC | 85.66 | 0.154721 | 7.25614 |
| FeCl$_3$@BC | 72.68 | 0.154301 | 8.06908 |
| Fe(NO$_3$)$_3$@BC | 122.56 | 0.144292 | 4.92112 |

a: BJH Desorption cumulative volume of pores between 1.7000 and 300.0000 nm width.

The SEM micrographs (Figure 2) revealed that the pore surfaces of FeCl$_3$@BC and Fe(NO$_3$)$_3$@BC were more intact as compared to that of BC. It might be due to iron oxide embedded in the biochar, which can effectively delay the biochar fracture [18]. It was also observed by a research group that owing to the development of iron oxide on the biochar surface, the carbon (C) content of biochar has decreased, and the O and Fe contents were increased after Fe$^{3+}$ modification [21]. In the present research same results by EDS analysis of C, O, Fe content have been observed. Corresponding EDS analysis demonstrated that Fe elements were detected in FeCl$_3$@BC and Fe(NO$_3$)$_3$@BC, indicating that iron oxides had

been successfully loaded onto FeCl$_3$@BC and Fe(NO$_3$)$_3$@BC, which could provide abundant active sites, thereby improving the elimination capacity of Cr(VI) [14]. Further, it was observed in Figure 2c that iron oxide particles are partially aggregated in the Fe(NO$_3$)$_3$@BC pores, and some of the iron oxide particles in the biochar may not contribute to the removal of Cr(VI) due to the limited opportunities for contact with Cr(VI) [26]. It further explains the reason that although the iron content of Fe(NO$_3$)$_3$@BC is higher than that of FeCl$_3$@BC, the Cr(VI) removal ability of Fe(NO$_3$)$_3$@BC is poor than that of FeCl$_3$@BC in the Cr(VI) removal experiment.

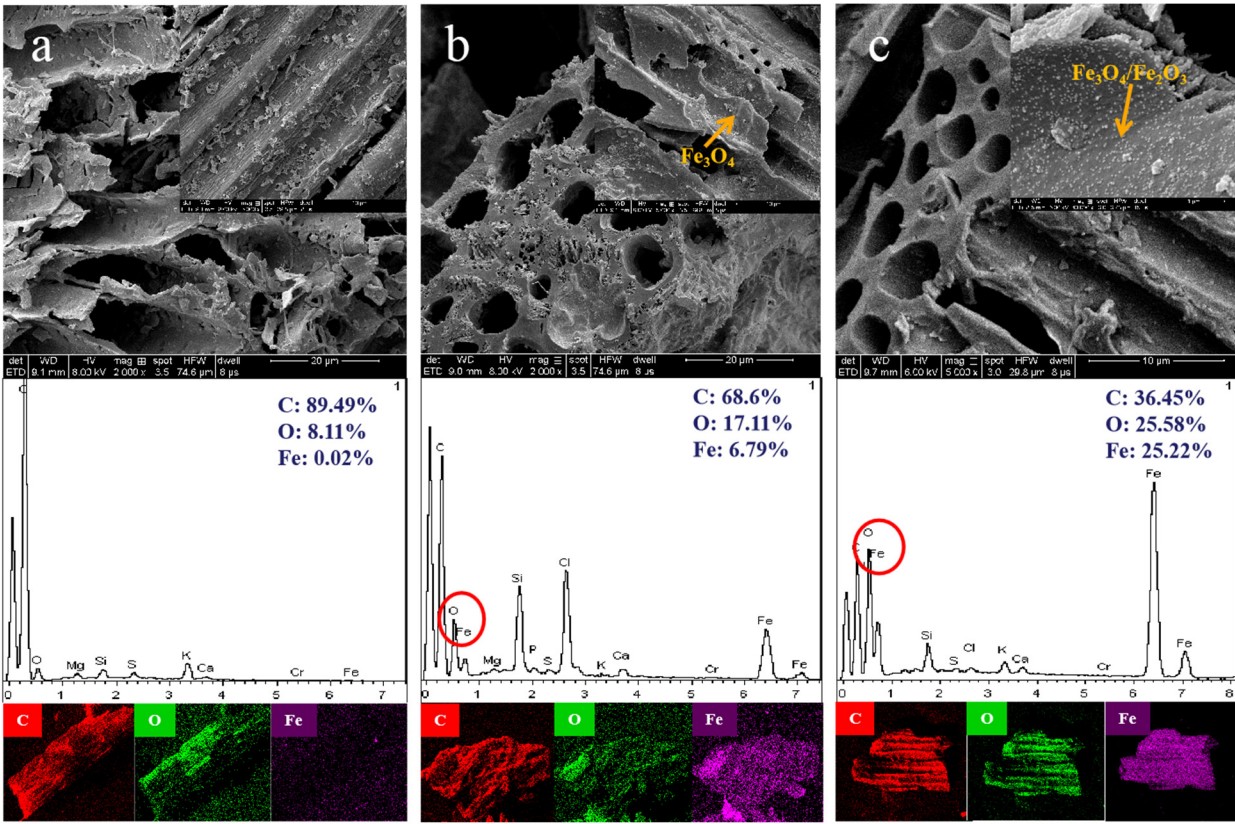

**Figure 2.** SEM-EDS analyses of BC (**a**), FeCl$_3$@BC (**b**), and Fe(NO$_3$)$_3$@BC (**c**).

The FTIR spectrum (Figure 3a) demonstrated a wide peak of adsorption at 3000–3700 cm$^{-1}$, related to stretching of -OH group of alcohols or phenols, in BC [14,27]. Moreover, the peak at 2920 cm$^{-1}$ was ascribed to the C-H vibration and peak at 2850 cm$^{-1}$ was due to CH$_x$ vibrations of aliphatic groups [2,28]. The peak at 1580 cm$^{-1}$ might be attributed to C=C and C=O stretching in the hydroxyl and lactone groups [29]. The distinctive peak at 1420 cm$^{-1}$ stated the presence of carboxylate group (O=C-O) [30] and 1107 cm$^{-1}$ were ascribed to the stretching of C-O [31,32]. The peaks between 700 and 900 cm$^{-1}$ were linked to C-H bending vibrations of aromatic group [33]. The functional groups such as -OH, -COOH, and -C-O could offer reaction sites for Cr(VI) [14].

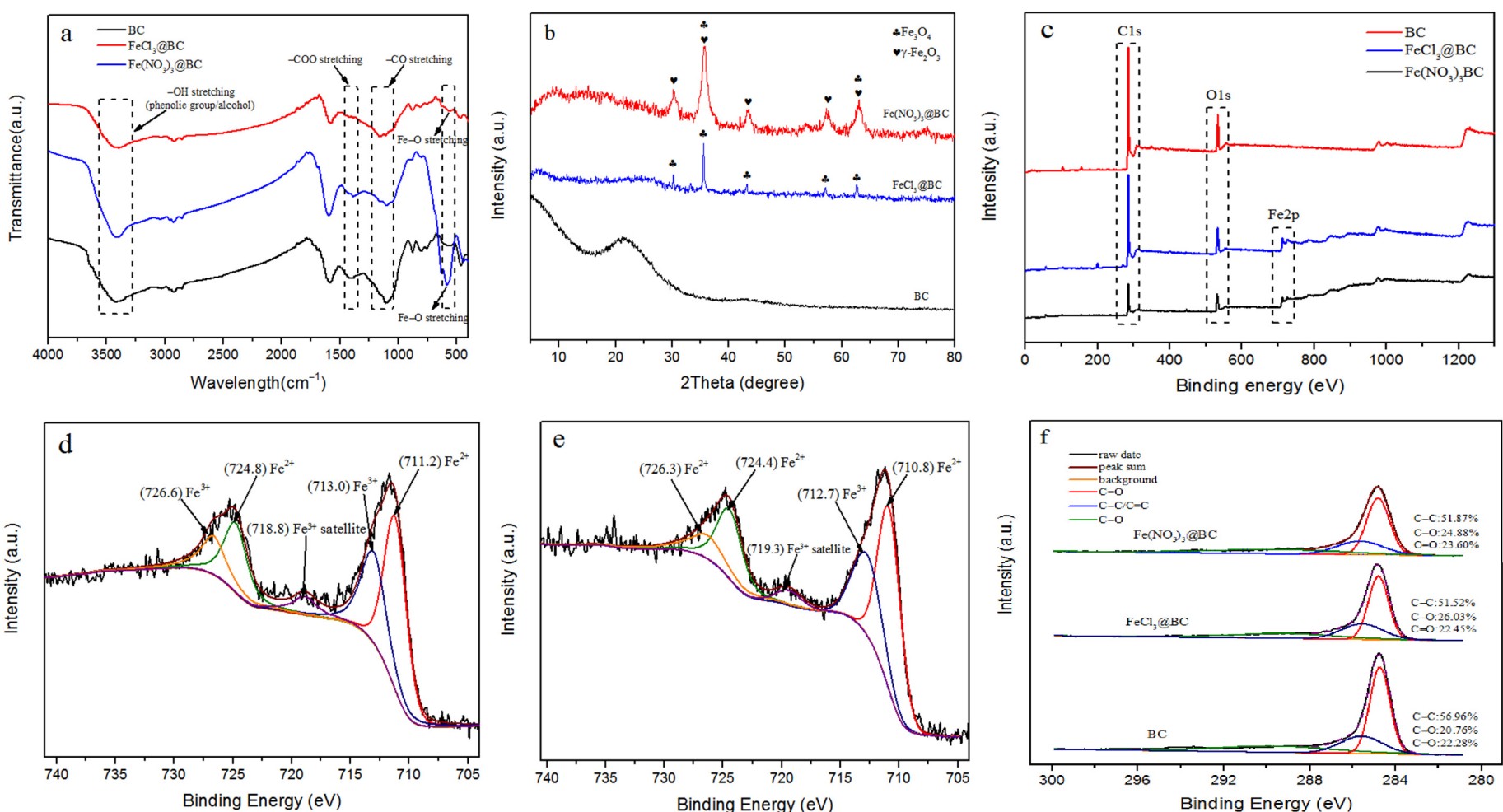

**Figure 3.** The comparative spectra of BC of FeCl₃@BC and Fe(NO₃)₃@BC as revealed by FTIR (**a**), XRD (**b**) and XPS (**c**). XPS Fe2p spectra of FeCl₃@BC (**d**) and XPS Fe 2p spectra of Fe(NO₃)₃@BC (**e**), XPS C1s spectra of BC, of FeCl₃@BC and Fe(NO₃)₃@BC (**f**).

The outcomes of $FeCl_3$@BC and $Fe(NO_3)_3$@BC were significantly different after $Fe^{3+}$ modification. The peaks of $FeCl_3$@BC disappeared at 1420 $cm^{-1}$, weakened at 3430 $cm^{-1}$, and also moved toward 1156 at 1107 $cm^{-1}$. The peak of $Fe(NO_3)_3$@BC was enhanced at 3430 and 1580 $cm^{-1}$. This might be attributed to loaded Fe on $FeCl_3$@BC or $Fe(NO_3)_3$@BC with functional groups (-OH, -C=O and -COOH) to form $-(COO)_n$-Fe, $-(CO)_n$-Fe, and $-O_n$-Fe [34]. In addition, a new absorption peak of $FeCl_3$@BC and $Fe(NO_3)_3$@BC appeared at 569 and 578 $cm^{-1}$, respectively, which can be linked to stretch vibration of Fe-O, signifying iron oxide loading on biochar [19,20,35].

In XRD spectrum (Figure 3b), it was found that the broad diffraction peak of BC at 21.42° indicated that there was a graphite base layer in the biochar [36]. The diffraction peaks of $FeCl_3$@BC at 30.20°, 35.52°, 43.15°, 57.04°, and 62.61° corresponded to the plane 220, 311, 400, 511, and 440 of $Fe_3O_4$ cubic crystals [37]. The diffraction peaks of $Fe(NO_3)_3$@BC at 30.25°, 35.52°, 43.15°, 57.46°, and 62.61° corresponded to $\gamma$-$Fe_2O_3$. In particular, the peaks at 35.52° and 62.61° showed the existence of $Fe_3O_4$. Therefore, the $Fe_3O_4$ distinctive peak intensity of $FeCl_3$@BC was obviously stronger as compared to BC and $Fe(NO_3)_3$@BC. However, different $Fe^{3+}$ modification techniques affected the morphology of iron oxides in materials. The conversion of $FeCl_3$ to $Fe_3O_4$ could be explained by Equations (7)–(10). The conversion of $Fe(NO_3)_3$ to $Fe_2O_3$ could be explained by Equation (11) [19,38,39]. In conclusion, these iron oxides were given during pyrolysis or co-precipitation by $Fe^{3+}$, and perform a significant role in Cr(VI) reduction. In addition, $FeCl_3$@BC has a superior removal effect of Cr(VI), which indicates that $Fe_3O_4$ is the key active component for the reduction of Cr(VI) to Cr(III).

$$Fe^{3+} + 3H_2O \rightarrow Fe(OH)_3 + 3H^+ \tag{7}$$

$$Fe(OH)_3 \rightarrow FeO(OH) + H_2O \tag{8}$$

$$C + H_2O \rightarrow CO\uparrow + H_2\uparrow \tag{9}$$

$$6FeO(OH) + 4CO \rightarrow 2Fe_3O_4 + 4CO_2\uparrow \tag{10}$$

$$2Fe(OH)_3 \rightarrow 2FeO(OH) + 2H_2O \rightarrow Fe_2O_3 + 2H_2O \tag{11}$$

To further discover the valence state of iron in the Fe-modified biochar, XPS spectroscopy was applied. In XPS spectra (Figure 3c), Fe peaks were detected in both $FeCl_3$@BC and $Fe(NO_3)_3$@BC. Among the Fe2p peaks of $FeCl_3$@BC (Figure 3d), the peaks of 711.2 and 724.8 eV combined energies linked to $Fe^{2+}$ in $Fe_3O_4$, while the peaks of 713 and 726.6 eV associated to $Fe^{3+}$, and there was a Fe satellite peak near 718.8 eV. Among the Fe2p peaks of $Fe(NO_3)_3$@BC (Figure 3e), the peaks of 710.8 and 724.4 eV were related to $Fe^{2+}$. Further, the peaks that corresponded to 712.7 and 726.3 eV were linked to $Fe^{3+}$, and there was a satellite peak of $Fe^{3+}$ near 719.3 eV [40,41]. It can be found from the C1s spectrum (Figure 3f) that with the introduction of Fe element, the content of C-O increased from 20.76 to 26.03% ($FeCl_3$@BC) and 24.88% ($Fe(NO_3)_3$@BC), the content of C=O increased from 22.28 to 22.45% ($FeCl_3$@BC) and 23.6% ($Fe(NO_3)_3$@BC), whereas the content of C=C decreased from 56.96 to 51.52% ($FeCl_3$@BC) and 51.87% ($Fe(NO_3)_3$@BC). It is speculated that $Fe^{3+}$ undergoes cation–$\pi$ interaction through C=C bond, and then bidentate chelate with C-O to form C-O-Fe bond, which is broken through electron transfer during pyrolysis to form iron oxide, thereby changing the carbon structure [38]. The results further explain the reasons for the structural changes and formation of iron oxides of Fe-modified biochar in XRD analysis. By analyzing the XPS spectra of $FeCl_3$@BC and $Fe(NO_3)_3$@BC, it could be observed that the selection of $Fe^{3+}$ solution had a direct effect on the $Fe^{2+}$ and $Fe^{3+}$ peak area (Table 2). Combined with XRD spectrum analysis, $Fe^{2+}$ mainly comes from $Fe_3O_4$. The peak area of $Fe^{2+}$ in $FeCl_3$@BC was 22921.951, while the peak area of $Fe^{2+}$ in $Fe(NO_3)_3$@BC was 16,800.615. It could be seen that $FeCl_3$@BC has a higher $Fe^{2+}$ content, which made $FeCl_3$@BC have better Cr(VI) removal ability. It indicated that $Fe_3O_4$ is the key active ingredient in the reduction of Cr(VI) to Cr(III).

**Table 2.** Peak area of $Fe^{2+}$ and $Fe^{3+}$ in $FeCl_3$@BC and $Fe(NO_3)_3$@BC.

| Material | $Fe^{2+}$ Area | $Fe^{3+}$ Area | Satellite Peak Area of |
|---|---|---|---|
| $FeCl_3$@BC | 22,921.951 | 15,773.833 | 2519.754 |
| $Fe(NO_3)_3$@BC | 16,800.615 | 8094.037 | 1333.63 |

Through the modification of $Fe^{3+}$, the zeta potential of biochar can be increased (Figure S3). It may be due to the combination of $Fe^{3+}$ and biochar, which introduces new functional groups like Fe-O and Fe-OH, thus leading to the development of biochar surface as more positively charged [42]. This can provide biochar with the ability to remove anionic Cr(VI) through electrostatic attraction. However, it was found that the zero potential of $FeCl_3$@BC is higher than BC and $Fe(NO_3)_3$@BC, which may be part of the reason why $FeCl_3$@BC had significantly increased the Cr(VI)-removal ability. Based on the above systematic characterization analysis, compared with $Fe(NO_3)_3$@BC, $FeCl_3$@BC has larger pore size, higher zeta potential, and higher content of $Fe_3O_4$, so it has better Cr(VI) removal ability. In conclusion, $FeCl_3$ is an excellent choice when choosing iron salt-modified biochar. Therefore, it is necessary to study the reaction process of Cr(VI) removal by $FeCl_3$-modified biochar in order to reveal its removal mechanism.

### 3.3. Analysis of the Influencing Factors FeCl₃@BC on Cr(VI) Elimination Performance

The biochar concentration dosage affect the number of active sites, which in turn affects the Cr(VI) elimination competence. It was detected that as the dosage of $FeCl_3$@BC increased, the material adsorption capacity for Cr(VI) has decreased (Figure 4a). It was due to increase of effective binding sites with rise in concentration of dosage, resulting in a diminution in the utilization efficiency of unit active site. Thus, considering the adsorption capability of $FeCl_3$@BC on Cr(VI), the follow-up experiments were conducted with 2 g/L of dosage.

The influence of pH on the Cr(VI) removal with $FeCl_3$@BC is clearly illustrated in Figure 4b. It was observed that increase in the solution initial pH negatively affects the Cr(VI) elimination. Generally, initial pH can affect $FeCl_3$@BC surface zero potential and the form of Cr(VI). The value of $pH_{pzc}$ of $FeCl_3$@BC was 7.03 (pH < $pH_{pzc}$), the positively charged surface of $FeCl_3$@BC was favorable for the removal of negatively charged chromium ions. However, when the pH > $pH_{pzc}$, the $FeCl_3$@BC surface was negatively charged, and it repelled negatively charged Cr(VI), thus not suitable for Cr(VI) removal. Further, the Cr(VI) species distribution was also monitored by pH, and thus affects the Cr(VI) removal efficiency [43]. Additionally, $HCrO_4^-$ was the prevailing species at pH 3.0–5.0, while its concentration decreased and its form changed to $CrO_4^{2-}$ with further rise in pH. Further, when the initial pH > 8.0, the Cr(VI) only occurred in the form of $CrO_4^{2-}$. $HCrO_4^-$ was favorably adsorbed in all forms of Cr(VI), owing to its low free energy of adsorption [44], which supports the greater removal capacity of Cr(VI) at an initial pH of 3.

Moreover, Figure 4c illustrated the effect of Cr(VI) initial amount (60 to 250 mg/L range) on the Cr(VI) removal. It was observed that the adsorption capacity of Cr(VI) decreases as the initial concentration of Cr (VI) increases. It might be owing to the reason that with increase in Cr(VI) initial concentration, the interaction probability of Cr(VI) with active surface sites of $FeCl_3$@BC and the driving force of Cr(VI) from the solution to the surface of the adsorbent increase [22]. Further, in groundwater environment, Cr(VI) often coexists with $NO_3^-$, $SO_4^{2-}$, $HCO_3^-$, and $PO_4^{3-}$ anions, so these ions often compete for adsorption sites with Cr(VI), and thus affect Cr(VI) elimination by $FeCl_3$@BC (Figure 4d). Among coexisting anions, $SO_4^{2-}$ and $PO_4^{3-}$ have the greatest influence on the Cr(VI) deletion. $HCO_3^-$ and $NO_3^-$ have no significant effect on Cr(VI) removal, which might be associated to the strong adsorption of $SO_4^{2-}$ and $PO_4^{3-}$ by $FeCl_3$@BC. This indicated that electrostatic adsorption has a noteworthy role in the elimination of Cr(VI) by $FeCl_3$@BC. Further, $PO_4^{3-}$ may also react with iron oxides, forming iron-phosphorus-complexed

on FeCl$_3$@BC surface, thus blocking adsorption sites, and influencing the removal of Cr(VI) [26].

It was further observed that adsorption kinetics provided important insights into the effectiveness and mechanism of the adsorption process [45,46]. The adsorption capacity of FeCl$_3$@BC on Cr(VI) has improved with an increase in adsorption time, and has reached to the equilibrium after 72 h (Figure 4e). Table 3 listed the relevant parameters of the pseudo-first-order, pseudo-second-order, and Elovich models of FeCl$_3$@BC adsorption of Cr(VI) at different initial concentrations. Elovich model had a higher R$^2$ value, thus indicated the elimination of Cr(VI) by FeCl$_3$@BC involves a series of adsorption processes such as diffusion in the solution phase or interface, surface activation [2,47]. Further, the $\alpha$ value higher than the $\beta$ value may reflect the early rapid adsorption [48].

**Table 3.** Adsorption kinetics and adsorption isotherm parameters.

| Biochar | C$_0$(mg/L) | Adsorption Kinetics | | | | | | | | | |
| --- | --- | --- | --- | --- | --- | --- | --- | --- | --- | --- | --- |
| | | $\ln(q_e - q_t) = \ln q_e - K_1 t$ | | | | $q_t = \frac{q_e^2 K_2 t}{1+q_e K_2 t}$ | | | $q_t = \frac{1}{\beta}\ln(1+\alpha\beta t)$ | | |
| | | $q_{e,exp}$ | $q_{e,cal}$ | $K_1$ | $R^2$ | $q_{e,cal}$ | $K_2$ | $R^2$ | $\alpha$ | $\beta$ | $R^2$ |
| | 60 | 26.26 | 21.86 | 0.65 | 0.766 | 23.45 | 0.038 | 0.884 | 154.15 | 0.32 | 0.99 |
| | 100 | 29.96 | 24.43 | 0.413 | 0.72 | 26.35 | 0.024 | 0.85 | 105.91 | 0.27 | 0.98 |
| FeCl$_3$@BC | 150 | 32.54 | 27.12 | 0.44 | 0.75 | 29.09 | 0.023 | 0.87 | 107.54 | 0.24 | 0.98 |
| | 200 | 33.18 | 28.00 | 0.52 | 0.76 | 29.92 | 0.026 | 0.883 | 158.2 | 0.25 | 0.98 |
| | 250 | 34.09 | 28.85 | 0.516 | 0.74 | 30.86 | 0.025 | 0.864 | 84.251 | 0.236 | 0.98 |
| Biochar | | Adsorption Isotherm | | | | | | | | | |
| | | Freundlich: $q_e = K_F C_e^{1/n}$ | | | | Langmuir: $q_e = \frac{K_L q_{max} C_e}{1+K_L C_e}$ | | | | | |
| | | $q_{max,exp}$ | $K_F$ | n | $R^2$ | $q_{max,cal}$ | | $K_L$ | | $R^2$ | |
| FeCl$_3$@BC | | 34.09 | 22.24 | 12.11 | 0.999 | 33.41 | | 0.46 | | 0.993 | |

For clear understanding of the adsorption of Cr(VI) from solution to FeCl$_3$@BC, the intra-particle diffusion model was applied for further analysis. Figure 4f illustrated that the intra-particle diffusion model has a three-stage adsorption process: The first stage k$_1$ was higher, indicating that Cr(VI) adsorption was a central-surface or intra-particle diffusion reaction [49]; the second stage was the internal diffusion stage (k$_1$ > k$_2$), which showed that the electrostatic adsorption and ion exchange mechanism on the surface participate in the elimination of Cr(VI), making it difficult for Cr(VI) to enter FeCl$_3$@BC [50,51]; while the third stage was the equilibrium phase of the adsorption process [42,52]. Meanwhile, it was worth noticing that the straight lines of the adsorption model of Cr(VI) did not cross the origin, thus specifying that the Cr(VI) removal process is controlled by diffusion, and not only by the rate control step [2,47]. Figure 4g illustrates the Cr(VI) adsorption isotherm on FeCl$_3$@BC with the optimum elimination capacity of 34.1 mg/g. At a lower concentration (7.48 mg/L), FeCl$_3$@BC also has a sturdy removal rate of Cr(VI) (26.3 mg/g). It showed that FeCl$_3$@BC was an effective adsorbent and had excellent Cr(VI) removal in a wide range of concentration. The adsorption isotherm date in Table 4 showed that the Freundlich model has the highest R$^2$ value, which indicated that there were non-uniform distributed adsorption centers on FeCl$_3$@BC surface and also Cr(VI) possess chemical adsorption properties [2,21]. Nevertheless, the maximum adsorption capability of FeCl$_3$@BC for Cr(VI) calculated by the Langmuir model was 33.41 mg/g, which exceeded other capacities reported in the literature (Table 4).

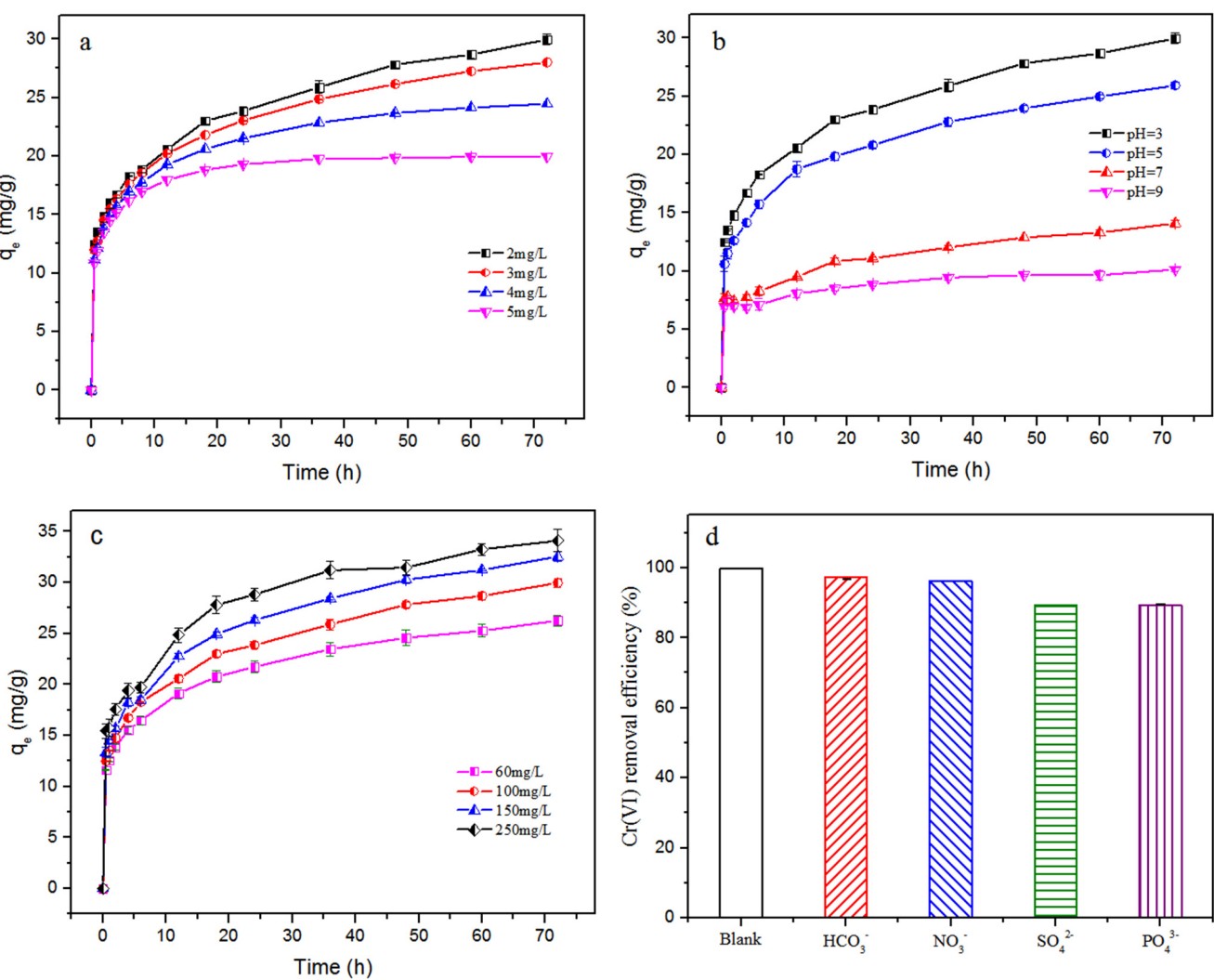

**Figure 4.** *Cont.*

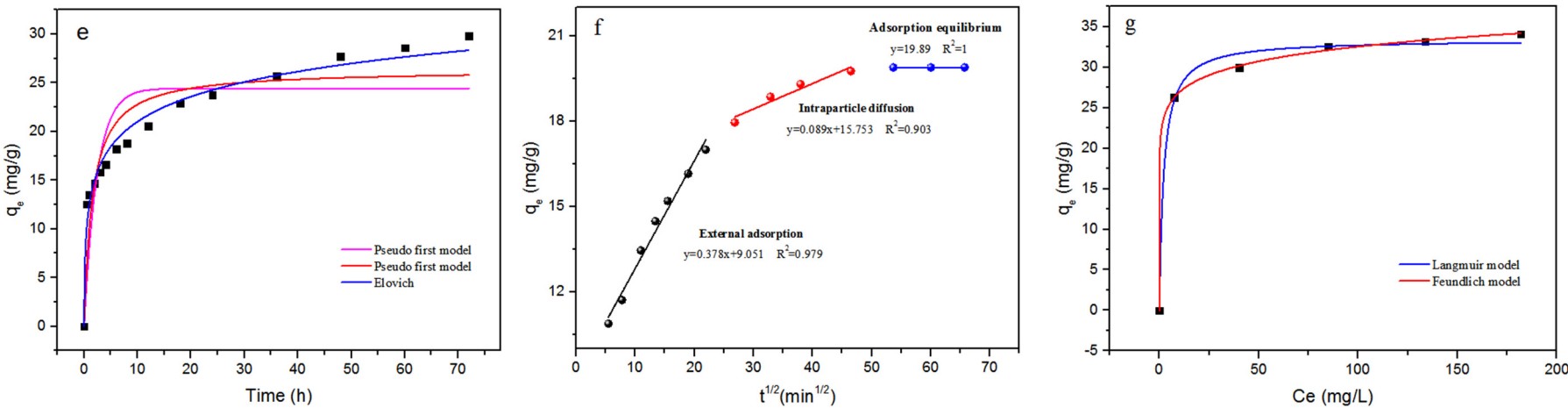

**Figure 4.** The effect of dosage of FeCl$_3$@BC on Cr(VI) elimination efficiency (**a**); influence of initial pH (**b**); removal efficiency of FeCl$_3$@BC at different initial concentrations of Cr(VI) (**c**); influence of competing ions (**e**); adsorption kinetics (**d**); intragranular diffusion model (dosage of 5 g/L) (**f**); and adsorption isotherm (**g**).

**Table 4.** Evaluation of adsorption capacities of different biochar for Cr(VI).

| Biochar | pH | $q_{max}$(mg/g) | Reference |
|---|---|---|---|
| $Fe_3O_4$-biochar | 3 | 8.35 | [25] |
| $\alpha$-$Fe_2O_3$-biochar | 5 | 19.51 | [53] |
| $Fe_3O_4$-biochar | 4.4 | 18.93 | [54] |
| $Fe_3O_4$-biochar | 2 | 23.85 | [55] |
| $\gamma$-$Fe_2O_3$-biochar | 3 | 11.25 | [56] |
| magnetic biochar | 1 | 27.2 | [46] |
| MMABC | 3 | 25.2 | [20] |
| FeBC-1 | 7 | 27.75 | [22] |
| $FeCl_3$@BC | 3 | 33.41 | This work |

Stability Analysis

After biochar adsorbs Cr(VI), desorption may occur, causing secondary pollution. Therefore, HCl (0.1 mol/L), NaCl (0.1 mol/L), and NaOH (0.1 mol/L) solutions were selected for elution experiments to explore the stability of $FeCl_3$@BC. The contents of Cr(VI) and Cr(III) in the solution after elution were demonstrated in Figure S4a. The results showed that the different properties of the eluent affect the elution volume of Cr(VI) and Cr(III). The rate of elution in a strong basic environment reached 6.9%, which was higher than other eluates. It might be due to the negative charge on $FeCl_3$@BC surfaces at higher pH, which boosted the electrostatic repulsion and reduced the electrostatic adsorption capability of Cr(VI) anions. Meanwhile, the total iron concentration was also tested in Figure S4b. The total iron content leached even under strong acid conditions was very low, which might be considered as an additional proof of $FeCl_3$@BC stability. Moreover, in a strong alkaline condition the elution rate was still low thus indicating that electrostatic adsorption was not the chief cause of Cr (VI) removal [22].

*3.4. Mechanism of Cr(VI) Elimination*

The morphology of Cr was monitored during reaction in water (Figure S5a). Initially, the solution only contained Cr(VI) ($T_{Cr}$ = Cr (VI) = 100 mg/L), but after addition of $FeCl_3$@BC, the concentration of both $T_{Cr}$ and Cr(VI) decreased significantly. The change in the concentration of both $T_{Cr}$ and Cr(VI) revealed that some portion of Cr(VI) has undergone reduction to Cr(III) during the reaction. As the experiment was carried out at pH 3, the $FeCl_3$@BC surface was positively charged and these conditions were favorable for the adsorption of Cr(VI), but not for Cr(III) adsorption [56]. At the beginning of the reaction, $FeCl_3$@BC has high reactivity to Cr(VI) and released dissolved Fe and Cr(III) (Figure S5b). But later on, the dissolved Fe concentration showed fluctuation, which might be owing to the Cr(III) and $Fe^{2+}/Fe^{3+}$ precipitation [57].

The SEM and EDS results of $FeCl_3$@BC before and after the reaction with Cr(VI) are shown in Figure 5. It can be found that the surface of $FeCl_3$@BC after the reaction becomes rough and a large amount of precipitation is observed. In addition, the atomic percent results of Cr provided by EDS have exposed that it is inferred that Cr(VI) and Cr(III) are adsorbed or precipitated on the surface of $FeCl_3$@BC. The XRD spectrum of $FeCl_3$@BC revealed a new peak of $FeCr_2O_4$ (Figure S6). This indicated that the reduced Cr(III) can be converted into Fe-O-Cr precipitation. It was also previously reported that stable $Cr_xFe_{1-x}(OH)_3$ crystals might be formed by reaction of Cr(III) with $Fe^{3+}$. However, the characteristic peak of $Cr_xFe_{1-x}(OH)_3$ was not observed in the XRD spectrum of $FeCl_3$@BC after the reaction, so the generation of amorphous $Cr_xFe_{1-x}(OH)_3$ was speculated in this reaction [25,26].

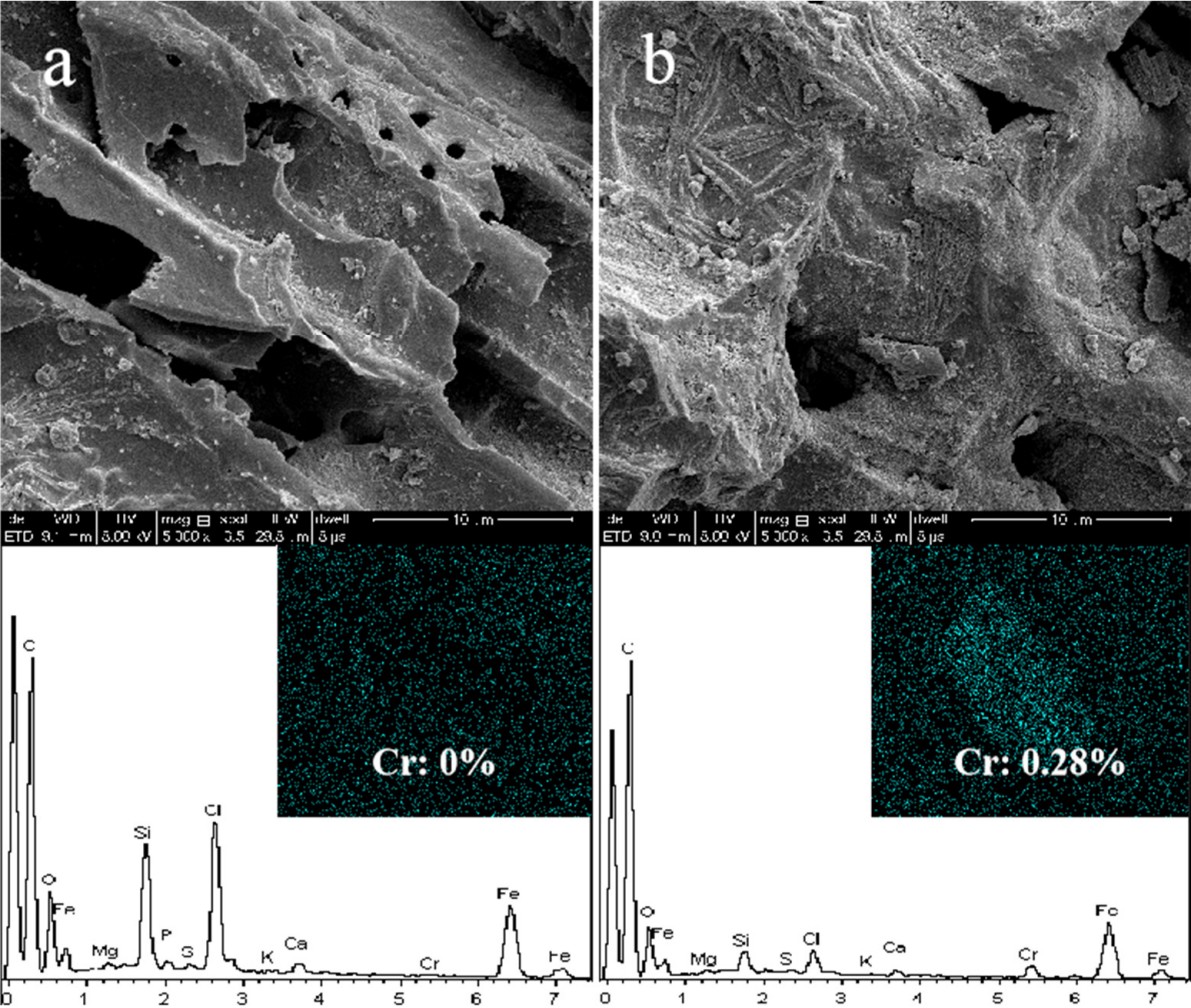

**Figure 5.** SEM-EDS analyses of FeCl$_3$@BC before reaction (**a**), after reaction (**b**).

To study the Cr(VI) elimination mechanism, XPS was applied for its characterization before and after the reaction. Thus, it was observed that the C1s peaks of FeCl$_3$@BC before and after the reaction were C-C/C=C at 284.8 eV, C-O at 285.8 eV, and C=O at 289.3 eV in Figure 6a [12,58–62]. After the reaction, the C-O content of FeCl$_3$@BC decreased from 26.03 to 21.06%, and the C=O content decreased from 22.45 to 22.45%, while the C-C/C=C content has increased from 51.52 to 59.14%. This indicated that C-O and C=O might be electron-donating groups in biochar, which could reduce Cr(VI) and might form C=C during electron transfer [38,63].

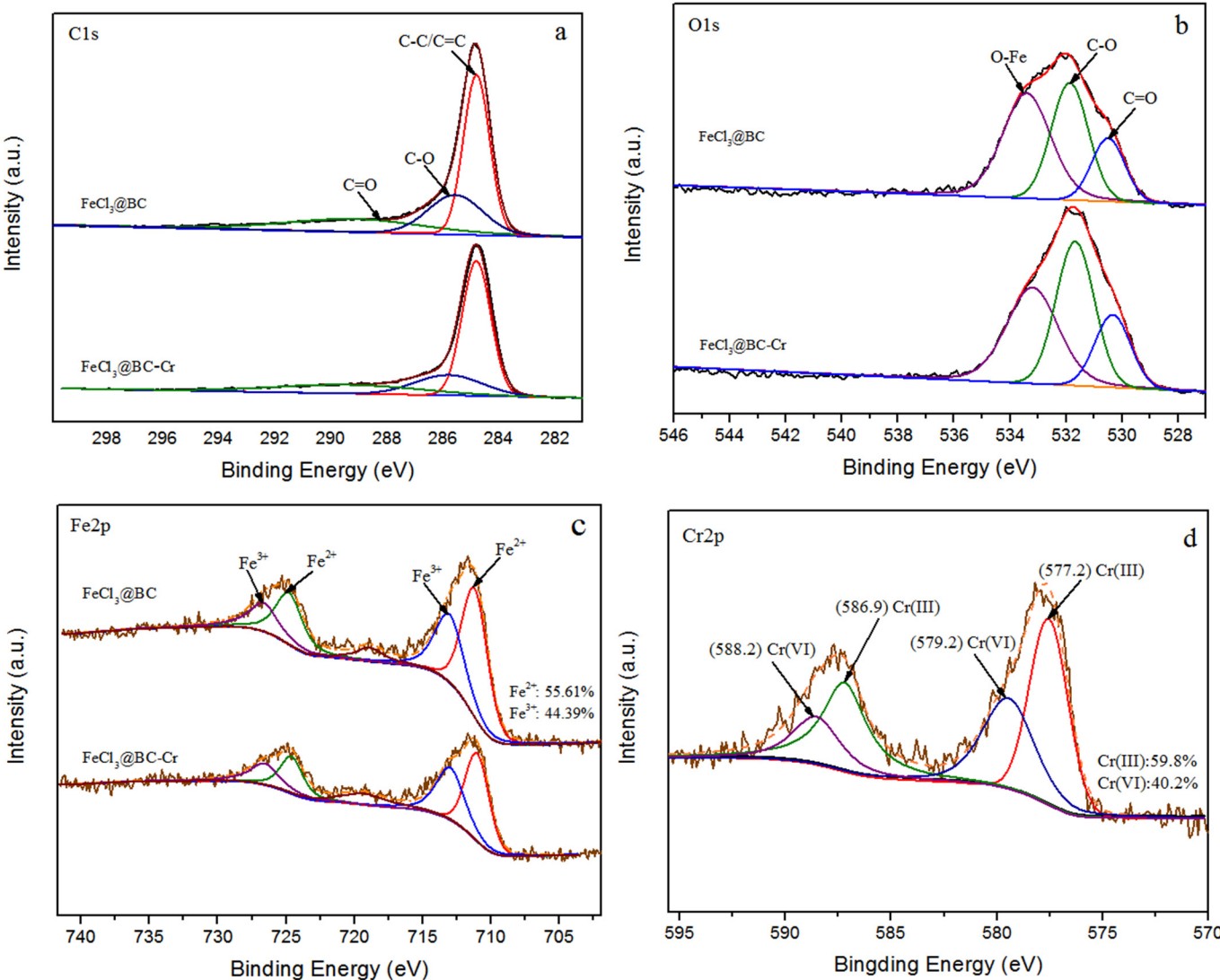

**Figure 6.** C1s spectra (**a**), O1s spectra (**b**), Fe2p spectra of FeCl$_3$@BC before and after the reaction (**c**), and Cr2p spectra of FeCl$_3$@BC after reaction (**d**).

The O1s peaks of FeCl$_3$@BC before the reaction were O-Fe at 530.5 eV, C-O at 531.8 eV, and C=O at 533.4 eV in Figure 6b [64–66]. While, after the reaction, the binding energy of FeCl$_3$@BC was slightly shifted to the low-energy region, thus indicating that the local binding environment has changed owing to Cr(VI) adsorption. In addition, Fe2p peak of FeCl$_3$@BC has also shown changes after the reaction in Figure 6c, demonstrating that Fe$^{2+}$ on the surface might be responsible for the reduction of Cr(VI) to Cr(III) [67]. Further, the Cr2p peak was considered to show the adsorption morphology of Cr(VI) on biochar in Figure 6d. After the reaction, the peaks of FeCl$_3$@BC at 579.2 and 588.2 eV corresponded to Cr(VI)(Cr2p3/2) and Cr(VI)(Cr2p1/2), while the peaks at 577.2 and 586.9 eV corresponded to Cr(III)(Cr2p3/2) and Cr(III)(Cr2p1/2) [12]. The Cr(VI) content was 40.2%, and the Cr(III) content was 59.8%. Thus, these findings demonstrated that the elimination of Cr(VI) is mainly its reduction to Cr(III).

Figure 7 and Table 5 summarize the possible mechanisms for FeCl$_3$@BC to remove Cr(VI): (1) absorption of Cr(VI) on FeCl$_3$@BC surface via porous adsorption and electrostatic attraction [37,68,69]; (2) Cr(VI) reduction to Cr(III) by gaining electrons from the oxygen-carrying groups on FeCl$_3$@BC surface, meanwhile, Fe$^{2+}$ on the surface could also convert Cr(VI) to Cr(III) [2]; (3) the Cr(III) produced by reduction may co-precipitate with Fe$^{3+}$ and Fe$^{2+}$ and be loaded on the biochar surface and pores [53,70].

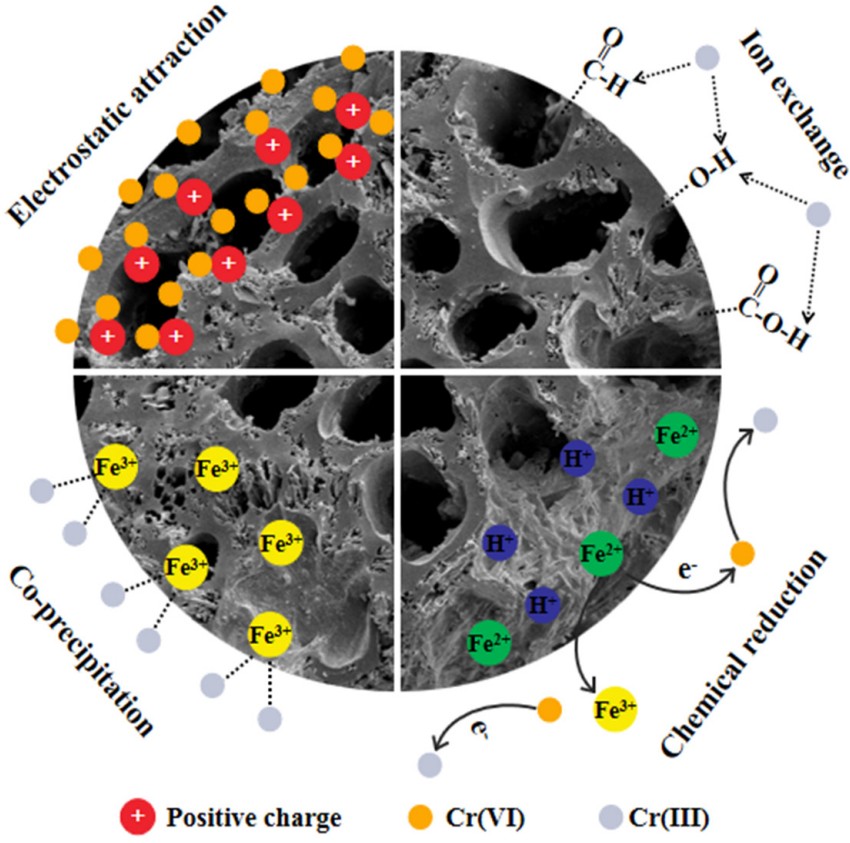

**Figure 7.** The mechanisms of hexavalent chromium Cr(VI) elimination by FeCl$_3$@BC.

**Table 5.** Adsorption mechanism of Cr(VI) via FeCl$_3$@BC.

| Reaction Mechanism | Procedure |
|---|---|
| Electrostatic adsorption | Fe-OH$^+$ + HCrO$_4^-$ /CrO$_4^{2-}$/Cr$_2$O$_7^{2-}$ → Fe-OH$^+$−HCrO$_4^-$ /CrO$_4^{2-}$/Cr$_2$O$_7^{2-}$ <br> FeCl$_3$@BC-OH$^+$ + HCrO$_4^-$ /CrO$_4^{2-}$/Cr$_2$O$_7^{2-}$ → FeCl$_3$@BC-OH$^+$−HCrO$_4^-$ /CrO$_4^{2-}$/Cr$_2$O$_7^{2-}$ |
| Chemical reduction | 3Fe$^{2+}$ + HCrO$_4^-$ + 7H$^+$ → 3Fe$^{3+}$ + 4H$_2$O + Cr$^{3+}$ <br> 3Fe$^{2+}$ + CrO$_4^{2-}$ + 8H$^+$ → 3Fe$^{3+}$ + 4H$_2$O + Cr$^{3+}$ <br> 6Fe$^{2+}$ + Cr$_2$O$_7^{2-}$ +14H$^+$ → 2Cr$^{3+}$ + 6Fe$^{3+}$ + 7H$_2$O <br> HCrO$_4^-$ + 7H$^+$+ 3e$^-$ → Cr$^{3+}$ + 4H$_2$O <br> CrO$_4^{2-}$+ 8H$^+$+ 3e$^-$ → Cr$^{3+}$ + 4H$_2$O |
| Ion exchange | 3Fe-OH$^+$+ Cr$^{3+}$ → 3Fe−O−Cr$^{3+}$ + 3H$^+$ <br> 3FeCl$_3$@BC−R−OH + Cr$^{3+}$ → 3(FeCl$_3$@BC−R−O)−Cr$^{3+}$ + 3H$^+$ <br> 3FeCl$_3$@BC−CHO + Cr$^{3+}$ → 3(FeCl$_3$@BC−C=O)−Cr$^{3+}$ + 3H$^+$ <br> 3FeCl$_3$@BC-COOH + Cr$^{3+}$ → 3(FeCl$_3$@BC-COO)−Cr$^{3+}$ + 3H$^+$ |
| Co-precipitation | (1−x)Fe$^{3+}$ + xCr$^{3+}$ + 3H$_2$O → Cr$_x$Fe$_{1-x}$(OH)$_3$ + 3H$^+$ <br> (1−x)Fe$^{3+}$ + xCr$^{3+}$ + 3OH$^-$ → Cr$_x$Fe$_{1-x}$(OH)$_3$ |

## 4. Conclusions

In this research, the removal of Cr(VI) and characterization on Fe-modified wheat straw biochar prepared by two different iron salt-modified methods have been explored. The experimental results indicated that the removal rate of FeCl$_3$@BC (99.87%) was significantly higher than that of Fe(NO$_3$)$_3$@BC (21.47%) and BC (15.13%). The characterization results have shown that FeCl$_3$@BC has a larger pore size, higher zeta potential, and higher iron oxide content than Fe(NO$_3$)$_3$@BC, and thus has a higher removal capacity of Cr(VI). Further, biochar modified with FeCl$_3$ has proved to promote the formation of Fe$_3$O$_4$ and thus has improved the reduction capacity of Cr(VI). In conclusion, FeCl$_3$ is an excellent choice when choosing iron salt-modified biochar. In addition, biochar modified by FeCl$_3$ is simple

and economical, so it is presenting a good application prospect in the removal of Cr(VI) pollution from groundwater.

**Supplementary Materials:** The following are available online at https://www.mdpi.com/article/10.3 390/w14060894/s1, Figure S1. (Schematic diagram of preparation of BC, $FeCl_3$@BC and $Fe(NO_3)_3$@BC). Figure S2. ($N_2$ adsorption-desorption isotherm for BC (**a**), $FeCl_3$@BC (**b**) and $Fe(NO_3)_3$@BC (**c**)). Figure S3. (Change of zeta potential of BC, $FeCl_3$@BC and $Fe(NO_3)_3$@BC). Figure S4. ((**a**)The Cr species content (Cr(VI) and Cr(III), respectively) in solution and distinct elution rates (the inset), (**b**) the leaching total Fe content after elution under three eluants with different acidity and alkalinity). Figure S5. ((**a**) The variance of TCr and Cr(VI) concentrations, (**b**) The variance of dissolved Fe concentration). Figure S6. (XRD spectra of before and after the reaction of $FeCl_3$@BC with Cr(VI)).

**Author Contributions:** (1) Y.J. and F.Y. made substantial contributions to the conception or design of the work; the acquisition, analysis, or interpretation of data; and the creation of new software used in the work; (2) I.A. and M.D. drafted the work or revised it critically for important intellectual content; (3) C.P. approved the version to be published; and (4) I.N., I.A. and C.P. agree to be accountable for all aspects of the work in ensuring that questions related to the accuracy or integrity of any part of the work are appropriately investigated and resolved. All authors have read and agreed to the published version of the manuscript.

**Funding:** This research was supported by the Guangdong College Students' Innovative Project (pdjh2021b0538, X201910580158), the Guangdong Provincial Key Laboratory of Environmental Health and Land Resource (2020B121201014) and the Technology Innovation Project of Zhaoqing (201904030103).

**Institutional Review Board Statement:** Not applicable.

**Informed Consent Statement:** Not applicable.

**Data Availability Statement:** Not applicable.

**Acknowledgments:** The authors are sincerely grateful to the testing center for providing the technical support.

**Conflicts of Interest:** Not applicable.

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
