# Peer review of "Remediation of Chromium (VI) from Groundwater by Metal-Based Biochar under Anaerobic Conditions"

_water, doi:10.3390/w14060894_

Round 1
Reviewer 1 Report
-The authors present an extensive and interesting paper conducted to clarify the specific effects of different iron modified biochars for the Cr(VI) removal. The manuscript has a proper structure and in general it is well written. In my opinion, the manuscript needs some work to become suitable for publication.
-It would strengthen the argument on the methodology if it would be clearly stated what is innovative about this specific methodological development.
-Keywords: Metal-based biochar; Cr(VI) remediation; groundwater; Fe(III); Pyrolysis; Cr(VI) 29 reduction and adsorption. Please change some keywords. Tittle and key words must not conntain the same words.
-I susggest to include some potos or diagram of the experiment
I wish those changes will contribute to improve your paper
Reviewer 2 Report
Dear authors
This work presents the results of the remediation of Chromium (VI) from Groundwater by Metal- based Biochar under Anaerobic Conditions: In my opinion is suitable for this journal.
You have put a lot of work in collecting and processing data. The results were presented under a well-structured frame and discussed satisfactory. It is an effort worthy of publication at this journal after some minor revision listed below :
Replace the title “Results”of section 3 with the “Results and discussion” one.
Figure 2: Please improve the analysis of spectra or enlarge these, if it is easy.
Conclusions : A conclusion about the tested Fe(ΝΟ3)3@BC lacks, please add.
I am looking forward to see this article published.
Kind regards
Reviewer 3 Report
The manuscript evaluated the specific effects of different iron modified biochars for the Cr(VI) removal. Overall, the approach of the study is good and could be useful in the public domain, but the manuscript needs considerable revision to reach the public domain. Authors are suggested to address following comments in order to make the manuscript suitable for publication.
* Abstract should be rewritten by detailing the aim and concept of the study. The abstract should state briefly the purpose of the study, the principal results and major conclusions.
*Provide significant words which are more relevant to the work in logical sequence as ‘keywords’. Also use keywords which are not present in title.
* Introduction is very general and need to be elaborative to explore the actual philosophy to design the experiment. The introduction is insufficient to provide the state of the art in the topic. The originality and novelty of the paper need to be further clarified. What progress against the most recent state-of-the-art similar studies was made in this study?
*The introduction of the paper must be extended and reformulated in order to provide a more comprehensive approach.
* Section 2.3; kindly cite the appropriate references properly.
* It would be necessary to develop more bioinformatic/statistical analyses in the present study.
* Section 3 should be Results and discussion.
*The manuscript does not provide interesting and technically sound discussion; it would be better to use more recent references in discussion.
*Under section, discussion, it is recommended to discuss and explain what should be the appropriate policies based on the findings of this study. Also, the results should be further elaborated to show how they could be used for real applications.
*Authors are suggested to draw major inferences/primary conclusions first quoting the data/results obtained followed by the secondary conclusions/ recommendations reached through the critical analysis/ investigation of the study. Based on the outcome of the study, the author(s) may recommend the extension of the present study as the future scope of research.
Author Response
Please the attachment

Round 2
Reviewer 3 Report
The authors addressed all the comments, therefore the manuscript may be accepted in the present form.